# Incidence of snakebites in Can Tho Municipality, Mekong Delta, South Vietnam—Evaluation of the responsible snake species and treatment of snakebite envenoming

**Vo Van Thang**[1], **Truong Quy Quoc Bao**[1], **Hoang Dinh Tuyen**[1], **Ralf Krumkamp**[2,3], **Le Hoang Hai**[4], **Nguyen Hai Dang**[4], **Cao Minh Chu**[4], **Joerg Blessmann**[1,2]*

**1** Institute for Community Health Research, College of Medicine and Pharmacy, Hue University, Hue, Vietnam, **2** Department of Infectious Disease Epidemiology, Bernhard Nocht Institute for Tropical Medicine, Hamburg, Germany, **3** German Center for Infection Research (DZIF), Hamburg – Lübeck – Borstel – Riems, Germany, **4** Department of Health, Can Tho City, Vietnam

* blessmann@bnitm.de

## Abstract

### Background

Data on incidence of snakebites and the responsible snake species are largely missing in Vietnam and comprehensive national guidelines for management of snakebite envenoming are not yet available. They are needed to estimate the scope of this health problem, to assess the demand for snake antivenom and to ensure the best possible treatment for snakebite victims.

### Methodology/Principle findings

A cross-sectional community-based survey was conducted from January to April 2018. Multistage cluster sampling was applied and snakebite incidence in Can Tho municipality, excluding two central districts of Can Tho city, was calculated at 48 (95%-confidence interval (CI): 20.5–99.8) snakebites per 100,000 person-years. Seven snakebite victims found during the survey reported 3 bites from green pit vipers and 4 bites from non-venomous snakes. In 2017 two treatment centres for snakebite envenoming in Can Tho city, the Military Hospital 121 and the Paediatric Hospital, received 520 admissions of snakebite victims. Two hundred sixty-seven came from Can Tho Municipality and 253 from neighbouring provinces. According to these data, the incidence of snakebites for Can Tho municipality was calculated at 21 (95%-CI: 18.5–23.7) snakebites per 100,000 person-years. Incidence was 14 (95%-CI: 12–17) snakebites per 100,000 person years in those 7 districts of the municipality which were part of the community survey. Green pit vipers were responsible for 92% of snakebite envenoming. Antivenom, antibiotics and corticosteroids were administered to 405 (90%), 379 (84%), and 310 (69%) out of 450 patients, respectively.

### Conclusions

Incidence of snakebites in Can Tho Municipality is relatively low and green pit vipers are responsible for the vast majority of bites. Approximately one third of snakebite patients

**Data Availability Statement:** All relevant data are within the manuscript and its Supporting Information files.

**Funding:** The study was funded by Else Kröner-Fresenius Stiftung, Grant Number: 2016_HA68. Website of funder: www.ekfs.de JB applied for and received the grant. The funders had no role in study design, data collection and analysis, decision to publish, or preparation of the manuscript.

**Competing interests:** The authors have declared that no competing interests exist.

sought medical care in hospitals and although hospital data still underestimate the real incidence of snakebites, these statistics are valuable and can be obtained fast and inexpensively. Evaluation of patients' records indicates the need for development of guidelines for management of snakebite envenoming in Vietnam to ensure a rational use of antivenom and ancillary treatments.

## Author summary

The World Health Organization included snakebite envenoming into the list of neglected tropical diseases in 2017 and launched a strategy for prevention and control of snakebite envenoming in 2019 in order to reduce the number of deaths and cases of disability by 50% before 2030. The neglect includes first and foremost epidemiology of snakebites, knowledge about management of snakebite envenoming and the medically relevant snake fauna, and access to life saving treatment with snake antivenom. In Vietnam reliable data on snakebite incidence and on snake species responsible for envenoming are largely missing. The present community-based survey showed that incidence of snakebites in Can Tho Municipality in the Mekong Delta in South Vietnam is significantly lower than in neighbouring countries and lower poverty rates among others are most likely the principal reason. Green pit vipers are responsible for the vast majority of snakebites. Treatment practices need to be reviewed and national guidelines are needed to achieve the best outcome for snakebite patients.

## Introduction

Snakebite envenoming is a neglected, poverty associated health problem in many developing countries with tropical and subtropical climate [1,2]. Only few publications addressing snakebite epidemiology and envenoming in Vietnam are published so far [3,4,5]. The World Health Organization (WHO) launched a strategy for prevention and control of snakebite envenoming in 2019, with the goal for all patients to have better overall care, so that the numbers of deaths and cases of disability are reduced by 50% before 2030 [6,7]. Four strategic aims will be pursued to achieve the goal, namely (i) to empower and engage communities (ii) to ensure safe and effective treatment, (iii) to strengthen health systems, and (iv) to increase partnerships, coordination and resources. Baseline data on incidence of snakebites and information on snake species responsible for envenoming in certain regions are needed to achieve these aims. Furthermore, well-communicated clinical guidelines on snakebite envenoming are essential to ensure safe and effective treatment, and rational use of antivenom, the precious essential drug. We conducted a community-based survey to investigate incidence of snakebites in Can Tho Municipality in the Mekong delta in South Vietnam. In addition, records of 520 patients with snakebite envenoming admitted to two treatment centres in Can Tho city were analysed, in order to calculate the incidence of snakebites in Can Tho Municipality on the basis of hospital documentation, to identify snake species responsible for envenoming and to assess treatment practices for snakebite victims.

## Methods

### Study design

A cross-sectional community-based survey with random multi-stage cluster sampling was performed between January and April 2018 in order to estimate the annual incidence of snakebites in Can Tho municipality, excluding two urban districts.

Records of 520 snakebite patients admitted to two snakebite treatment centres in 2017, namely the Military Hospital 121 and the Paediatric Hospital in Can Tho city, were reviewed to calculate snakebite incidence in Can Tho Municipality and to evaluate treatment practices applied for patients admitted with snakebite envenoming.

## Study site and study population

Can Tho Municipality covers an area of 1,439 km$^2$ and is divided into nine districts, namely Ninh Kieu, Binh Thuy, Cai Rang, O Mon, Thot Not, Phong Dien, Co Do, Vinh Thanh and Thoi Lai. These districts are again divided into 85 administrative units, referred to as communes, wards and towns. The municipality is located in the center of the Mekong Delta and borders the provinces of An Giang, Hau Giang, Kien Giang, Vinh Long, Dong Thap. It is the traffic hub of river, road and air routes with a population of 1,272,800 inhabitants in 2017 [8]. Seven districts were included in the study with a population of 881,800 inhabitants. The densely populated commercial center of Can Tho city, which covers most of Ninh Kieu and Binh Thuy districts with a population of 391,000 was excluded from the survey because it comprises mostly stores, hotels, restaurants, multi-storey urban buildings and industrial zones. Geography of the study region is characterized by lowland of the Mekong Delta with rice fields, fruit and vegetable plantations, which are crisscrossed by navigable waterways.

## Sample size and random selection

The number of snakebite incidents per year was estimated at 100 per 100,000 (0.1%) persons per year. This estimation was based on results from a recently published study on snakebite incidence from central Vietnam and data from other Asian countries [3]. Determining a precision of 5% and a confidence level of 95%, the estimated sample size to calculate single proportions was approximately 15,000 individuals [9].

Multi-stage random cluster sampling was applied and research randomizer version 4 was used to generate random numbers [10]. In the first stage, 25 (39%) out of 64 communes were randomly chosen. In each community, approximately 4.5% of the population were randomly selected to reach the envisaged sample size.

## Household and snakebite victim survey

The study team visited 3,747 households, with the support of trained local health workers. All people who lived in a household in 2017 were listed by age and gender. The chief of household or a representative was asked about an incident of snakebite that occurred in the family in 2017. We consider recall bias for snakebites in our study setup negligible. Those household members, who reported a snakebite were subjected to a second questionnaire about the circumstances of the bite, snake species, symptoms, the month in which the bite happened, treatment and outcome.

## Evaluation of hospital records

Records of 450 patients with snakebite envenoming admitted to the Military Hospital 121 (435 patients), and Paediatric Hospital (15 patients) in Can Tho city in 2017 were analysed for gender, age, place of residence, snake species responsible for the bite, clinical symptoms, lab results, treatment and outcome. For further 70 patients admitted to the Paediatric Hospital, only the place of residence was available. Based on these 520 records, incidence of snakebite was calculated for the entire Municipality with 9 districts, and separately for the two urban districts Ninh Kieu and Binh Thuy and the 7 other districts which were part of the community

survey. Snakebite patients admitted to district hospitals were not included in order to avoid overlapping. There were only very few cases and most of them were transferred to the central level.

## Statistical analysis

Epidata version 3.1 was used for data entry and Predictive Analytics Software (PASW) version 20 for statistical analysis [11]. Snakebite incidence was calculated as the number of snakebites per 100,000 persons per year, using the number of snakebite incidents found during the survey as nominator and the number of interviewees as denominator. The 95%-confidence interval (CI) for an incidence rate was calculated. Incidence of snakebites from hospital data was calculated with the number of snakebite patients with a residency in Can Tho Municipality admitted to two treatment centers in 2017 in relation to the population of the municipality. We assumed that all snakebite patients would seek treatment at the study hospitals.

## Ethical statement

The Institutional Ethics Committee of Hue University of Medicine and Pharmacy in Hue, Vietnam approved the study (Approval number H2018/10). Written informed consent was obtained from the head or representative of each household on behalf of his or her family.

# Results

A total of 14,445 individuals (7,251 females; 50,2%) from 3,747 households participated in the survey. From the initially randomly selected 16,497 individuals, 2,052 (12.4%) could not be interviewed, because families moved away or were not available. The median age of interviewees was 36 years (range 1–102).

## Annual incidence of snakebites and characteristics of snakebite victims

Seven of 14,445 interviewees reported a snakebite during 2017, and annual incidence is 48 per 100,000 persons per year (CI-95%: 20.5–99.8). Five of the snakebite victims were males and two females with a male/female ratio of 2.5:1 and the median age was 37 (range 11–70). Three times a bite from green pit viper and four times a bite from non-venomous snake species was reported. Two snakebites occurred during the rainy season between May and October and five snakebites during the dry season between November and April. All snakebites happened outdoor either in the garden or in the rice field. Four victims, including all patients with green pit viper bites received treatment in a hospital and the three victims who reported a bite from a non-venomous snake were treated by a traditional healer or self-treated at home. The outcome was favourable without sequelae for all seven snakebite victims.

## Evaluation of hospital records of snakebite patients

In 2017 a total of 520 snakebite patients were admitted to two snakebite treatment centers in Can Tho city, the Military Hospital 121 (435 patients) and the Paediatric Hospital (85 patients). Two hundred sixty-seven (51%) out of 520 patients were residents of Can Tho Municipality and 253 patients (49%) came from neighboring provinces. Giving a total population of 1,272,800 the incidence of snakebites in Can Tho Municipality was calculated at 21 snakebites per 100,000 persons per year (95%-CI: 18.5–23.7). There were 140 snakebite patients in the two urban districts Ninh Kieu and Binh Thuy and 127 in the other 7 districts, which were part of the community survey. Considering current population data, incidence of snakebites is calculated at 36 snakebites per 100,000 persons per year (95%-CI: 30–42) in the

**Table 1. Characteristics of 450 patients admitted to Military Hospital 121 and Paediatric Hospital in Can Tho city in 2017.**

| | | Green pit viper (n = 414) | Cobra (n = 5) | No ID (n = 31) | Total (n = 450) |
|---|---|---|---|---|---|
| **Gender** | | | | | |
| Male | | 261 (63%) | 5 (100%) | 21 (68%) | 287 (64%) |
| Female | | 153 (37%) | 0 (0%) | 10 (32%) | 163 (36%) |
| Male/female ratio | | 1.7 | n.a. | 2.1 | 1.8 |
| **Age** (years) | | | | | |
| Median | | 43 | 42 | 40 | 43 |
| IQR | | (32–54) | (36–48) | (22–49) | (31–54) |
| **Symptoms after the bite** | | | | | |
| Swelling | | 408 (99%) | 3 (60%) | 8 (26%) | 419 (93%) |
| Signs of bleeding | | 119 (29%) | 1 (20%) | 7 (23%) | 127 (28%) |
| Neurotoxic signs | | 3 (1%) | 0 (0%) | 0 (0%) | 3 (1%) |
| None | | 5 (1%) | 2 (40%) | 18 (58%) | 25 (6%) |
| **Laboratory tests** | | | | | |
| INR | <1.3 | 270 (65%) | 4 (80%) | 14 (45%) | 288 (64%) |
| | 1.3–2.0 | 12 (3%) | 0 (0%) | 1 (3%) | 13 (3%) |
| | >2 | 4 (1%) | 1 (20%) | 1 (3%) | 6 (1%) |
| | Not available | 128 (31%) | 0 (0%) | 15 (49%) | 143 (32%) |
| Platelet | ≥150,000/μl | 390 (94%) | 5 (100%) | 29 (94%) | 424 (94%) |
| | <150,000/μl | 23 (6%) | 0 (0%) | 1 (3%) | 24 (5%) |
| | Not available | 1 (<1%) | 0 (0%) | 1 (3%) | 2 (< 1%) |
| Platelet <150,000/μl and INR μ1.3 | | 3 | 0 | 0 | 3 |
| **Treatment** | | | | | |
| Snake antivenom | | 390 (94%) | 5 (100%) | 10 (32%) | 405 (90%) |
| Antibiotics | | 358 (86%) | 5 (100%) | 16 (52%) | 379 (84%) |
| Corticosteroids | | 293 (71%) | 3 (60%) | 14 (45%) | 310 (69%) |
| Intravenous fluids | | 390 (94%) | 5 (100%) | 13 (42%) | 408 (91%) |
| Analgesics | | 407 (98%) | 5 (100%) | 27 (87%) | 439 (98%) |
| **Outcome** | | | | | |
| No sequela | | 394 (95%) | 3 (60%) | 22 (71%) | 419 (93%) |
| Disability | | 0 (0%) | 0 (0%) | 0 (0%) | 0 (0%) |
| Death | | 0 (0%) | 0 (0%) | 0 (0%) | 0 (0%) |
| No information | | 20 (5%) | 2 (40%) | 9 (29%) | 31 (7%) |

Abbreviations: INR = International Normalized Ratio; ID = Identification

two urban districts and 14 snakebites per 100,000 persons per year (95%-CI: 12–17) in the 7 other districts.

A total of 450 patients' records were further evaluated, 15 from the Paediatric Hospital and 435 from the Military Hospital 121. Characteristics of these 450 patients, including laboratory findings, received treatment and outcome are outlined in Table 1. The male/female ratio was 1.8:1, with 287 males and 163 females. The median age was 43 (IQR 31–54). In 414 cases (92%) the snake had been identified as a green pit viper, in 5 cases (1%) as cobra and in 31 cases (7%) identification was not documented. Swelling at the bite site was recorded in 419 (93%), clinical signs of bleeding in 127 (28%) and neurotoxic signs in 3 (1%) cases. In 286 (69%) out of 414 patients bitten by a green pit viper the International Normalized Ratio (INR) and in 413 (99,8%) the platelet count was available. Sixteen (6%) patients had an INR >1.3 and 23 (6%) a platelet count <150,000/μl. In three patients both parameters were abnormal, which

confirmed a coagulation disorder for 36 (9%) out of 414 patients. In the green pit viper group 119 (29%) patients had clinical signs of bleeding. Platelet count was available for all 119 patients and INR for 90 (76%) patients. Only 3 (3%) out of these 119 patients had an INR >1.3 and/or platelets <150,000/μl.

In both hospitals, two monovalent snake antivenoms against venom of *Trimeresurus albolabris* and *Naja kaouthia* from the Institute of Vaccines and Medical Biologicals (IVAC) in Nha Trang, Khanh Hoa province, Vietnam was available. They have been administered to 405 (90%) out of 450 patients. All 5 patients after a cobra bite received monovalent *Naja kaouthia* antivenom and 390 (94%) out of 414 patients with a green pit viper bite received monovalent *Trimeresurus albolabris* antivenom. Most frequent medications given were Analgesics (n = 439; 98%), intravenous fluids (n = 408; 91%), antibiotics (n = 379; 84%) and corticosteroids (n = 310; 69%). For 31 (7%) patients outcome could not be determined, because they left the hospital against medical advice or were transferred to another hospital. No death or disability was recorded for the remaining 419 patients.

## Discussion

Incidence of snakebites in Can Tho Municipality, excluding two districts of Can Tho city centre, was calculated at 48 snakebites per 100,000 persons per year. This is a similar incidence found recently in the lowland region in Thua Thien Hue province in central Vietnam with 69 snakebites per 100,000 persons per year [3]. In comparison with data from Laos and Myanmar, where incidence was up to 1,105 and 116 per 100,000 persons per year, respectively, incidence rate in Can Tho Municipality is rather low [12,13]. Several reasons explain the relatively low risk for snakebite in this area. The Mekong delta is a very fertile and productive region in Vietnam and mechanisation of agriculture is well advanced. Preparation of rice fields, rice plantation and harvest are largely mechanized, which leads to significant risk reduction. The population density is very high and most of the land is used for rice and fruit plantations. Together with increasing residential and industrial areas, snakes are more and more deprived of their habitat, which results in a decreasing snake population. Furthermore, snake hunting for different purposes, such as food, snake wine, skin trade and medicinal healing practices, reduces the local cobra population [14]. The different snake fauna in Myanmar and Laos with Russell's vipers *(Daboia russelii)* predominately found in Myanmar and the Malayan pit vipers *(Calloselasma rhodostoma)* in Laos will also have a significant impact on incidence of snakebites. Ecology and development of the whole Mekong delta region are rather similar, and results of the present survey most likely apply also for other provinces in the river delta.

Most snakebite victims in Can Tho Municipality are referred to our two study hospitals because antivenom against the green pit viper (*Trimeresurus albolabris*) and the monocled cobra (*Naja kaouthia*) is available. Only few additional snakebite victims are treated in district hospitals and the General hospital of Can Tho city where antivenom is only in limited supply. On the basis of 267 admissions of snakebite patients to these study hospitals in 2017, incidence of snakebites in Can Tho municipality was calculated at 21 snakebites per 100,000 persons per year. Further differentiation of snakebite incidence from hospital records showed an incidence of 36 and 14 snakebites per 100,000 persons per year in the two urban districts Ninh Kieu and Binh Thuy excluded from the community survey and in the 7 districts included in the community survey, respectively. Thus, hospital records revealed approximately one third of snakebite incidence found in the community survey. However, taking into account that part of all bites happening in the community is from non-venomous snakes, which are more likely treated at home like minor wounds and green pit viper bites being usually less severe, still a significant number of patients with snakebite envenoming sought medical care in hospitals.

Evaluation of 450 patients' records showed that green pit vipers caused 414 (92%) out of 450 snakebites in Can Tho Muncipality. It is not known whether these are predominately *Trimeresurus albolabris* or other species of the genus *Trimeresurus* [15]. Bites from snakes of the genus *Trimeresurus* are usually less severe and rarely life-threatening. According to WHO guidelines for treatment of snakebite envenoming, patients with clinical signs of bleeding, laboratory signs of coagulation disorder and swelling of more than 50% of the bitten extremity should receive antivenom after a viper bite [16]. The present evaluation of 414 patients with green pit viper bite showed that less than 10% had laboratory signs of a coagulation disorder and only 3% of 119 patients with clinical signs of bleeding had laboratory parameters indicating coagulation disorder, but for 408 (99%) cases swelling at the bite site without further information about the extension was recorded. It indicates that the majority of cases received antivenom for treatment of local cytotoxic effects. The beneficial impact of antivenom to reverse local cytotoxic effects is arguable and well-designed clinical trials that prove a curative effect, even for extensive swellings, are missing and so far recommendations are based on expert opinions only. A recent study from the United States of America investigated the effect of antivenom on limb function in patients with copperhead snake envenoming. Limb disability was reduced 14 days after copperhead envenomation in the antivenom group, but the study was underpowered, because the target sample size could not be reached and after 30 days the outcome did not differ [17]. Snake antivenom is very precious, and supply is limited in Southeast Asia and should be reserved for those with an indication based on convincing evidence.

For ancillary treatment, 380 (84%) and 310 (69%) received antibiotics and corticosteroids, respectively. However, antibiotics are rarely needed after snakebites and corticosteroids are not recommended at all [16,18].

The evaluation of these 450 records indicates that there is a need to revise treatment schemes and to develop guidelines for management of snakebite envenoming in Vietnam, on the basis of existing WHO guidelines, to ensure rational use of antivenom and ancillary treatment.

The main drawback of the present study is the retrospective analysis of patients' data from hospital records which are not always accurately documented. Patient's residence documented in the hospital record might not be the current living address or correspond to the place where the snakebite happened. However, we believe that on the aggregated data this effect is minor. The overall low number of snakebite cases within the community (7 cases) effects the statistical incidence estimation, which increased the corresponding confidence interval.

In conclusion snakebite incidence in Can Tho Municipality is rather low and green pit vipers are responsible for the vast majority of bites. Although hospital statistics still underestimate the real number of cases, they allow a reasonable estimation of venomous snakebite incidence within the region. To improve management of snakebite victims, regular training of healthcare professionals on snakebite treatment should be conducted, the community members should be informed and a sustainable stock of effective antivenom has to be in place. Furthermore, national guidelines for treatment of snakebite envenoming need to be developed, in order to achieve the best outcome for patients and the health system in terms of medical care and economic efficiency.

## Supporting information

**S1 Checklist. STROBE statement.** Checklist of items for observational studies.
(DOC)

## Acknowledgments

We would like to express our thanks to Dang Trung Kien, Tran Hoang Thong, Dinh Quoc Thai, Nguyen Thanh Trinh, Huynh Van Vu, Nguyen Phu Gia for their support to collect patient's data at the Military Hospital 121 in Can Tho city. A special thank-you goes to Ms Nguyen Thi Thu Thuy, secretary at Institute for Community Health Research (ICHR) for financial planning of the surveys and coordination of our work. We are grateful to the staff of the children's hospital, district health offices, district hospitals and community health centres of Can Tho Municipality for their support.

## Author Contributions

**Conceptualization:** Vo Van Thang, Ralf Krumkamp, Joerg Blessmann.

**Data curation:** Truong Quy Quoc Bao, Hoang Dinh Tuyen, Le Hoang Hai, Nguyen Hai Dang, Joerg Blessmann.

**Formal analysis:** Vo Van Thang, Truong Quy Quoc Bao, Hoang Dinh Tuyen, Ralf Krumkamp, Joerg Blessmann.

**Funding acquisition:** Joerg Blessmann.

**Investigation:** Truong Quy Quoc Bao, Hoang Dinh Tuyen, Le Hoang Hai, Nguyen Hai Dang, Joerg Blessmann.

**Methodology:** Ralf Krumkamp.

**Project administration:** Vo Van Thang, Truong Quy Quoc Bao, Hoang Dinh Tuyen, Le Hoang Hai, Nguyen Hai Dang, Cao Minh Chu, Joerg Blessmann.

**Supervision:** Vo Van Thang, Cao Minh Chu, Joerg Blessmann.

**Writing – original draft:** Joerg Blessmann.

**Writing – review & editing:** Vo Van Thang, Truong Quy Quoc Bao, Hoang Dinh Tuyen, Ralf Krumkamp, Le Hoang Hai, Nguyen Hai Dang, Cao Minh Chu.

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
