## [Decision Letter · Decision Letter 0]

15 Apr 2020

Dear Dr Blessmann,

Thank you very much for submitting your manuscript "Incidence of snakebites in Can Tho Municipality, Mekong Delta, South Vietnam - Evaluation of the responsible snake species and treatment of snakebite envenoming" for consideration at PLOS Neglected Tropical Diseases. As with all papers reviewed by the journal, your manuscript was reviewed by members of the editorial board and by several independent reviewers. In light of the reviews (below this email), we would like to invite the resubmission of a significantly-revised version that takes into account the reviewers' comments, especially the comments and suggestions of Reviewer #2. 

We cannot make any decision about publication until we have seen the revised manuscript and your response to the reviewers' comments. Your revised manuscript is also likely to be sent to reviewers for further evaluation.

Sincerely,

Arunasalam Pathmeswaran

Guest Editor

Janaka de Silva

Deputy Editor

Reviewer's Responses to Questions

**Key Review Criteria Required for Acceptance?**

**Methods**

-Are the objectives of the study clearly articulated with a clear testable hypothesis stated?

-Is the study design appropriate to address the stated objectives?

-Is the population clearly described and appropriate for the hypothesis being tested?

-Is the sample size sufficient to ensure adequate power to address the hypothesis being tested?

-Were correct statistical analysis used to support conclusions?

-Are there concerns about ethical or regulatory requirements being met?

Reviewer #1: The aim of the study is clearly defined and is restricted to an area in South-Vietnam. The sample size is therefore limited, however, it is sufficient to answer the question to what extent snakebite causes health-problems. The statistical analysis has been correctly performed and there are no concerns about ethical standards.

Reviewer #2: The methods are well described and generally sound. The population under study is clearly defined and the sample size sufficient to reach conclusions on snakebites in the Can Tho municipality in Vietnam. However, in the estimation of incidence based on community surveys, the authors excluded two districts because being mostly urban settings. But in the estimation of incidence based on hospital records, the authors used the total population of the Can Tho Municipality, hence including the two districts that were excluded in the other estimation. This creates a difference in the way incidence was estimated and therefore complicates the comparison between the two data on incidence. This issue is acknowledged by the authors when discussing the limitations of the study. Would it be possible to identify the locale where bites occurred from the hospital records and therefore to estimate the incidence from hospital records excluding the patients from the two central districts? This would allow the comparison between community-based data and hospital-based data.

Reviewer #3: The method of community survey may be helpful in situation where the community may not be in favour of seeking treatment at from a hospital. This is therefore appropriate for the locality of this study. This will certainly a good support/supplement to the formal hospital admission data. However, overlapping of the same patient may occur and should be taken into account.

**Results**

-Does the analysis presented match the analysis plan?

-Are the results clearly and completely presented?

-Are the figures (Tables, Images) of sufficient quality for clarity?

Reviewer #1: There is only one table which summarizes the results of the study. The data evaluated are well presented and the results are clearly described.

Reviewer #2: In general, the presentation of results is clear. However, there are some limitations derived from the difficulties in getting precise information from hospital records: (1) The identification of the snakes responsible for the bites is always difficult when information is collected from hospital records. This is because the uncertainty in the identification of the snakes by patients or relatives. Since green snakes are relatively easy to recognize, this helps in the case of bites by Trimeresurus sp. The same occurs in the case of Naja sp bites. But the identification of non-venomous snakes is highly unreliable. I recommend only referring to non-venomous snakes without indicating the particular species or genera.

In line 220 and afterwards it is stated that “In both hospitals, snake antivenom against venom of Trimeresurus albolabris and Naja kaouthia from the Institute of Vaccines and Medical Biologicals (IVAC) in Nha

Trang, Khanh Hoa province, Vietnam was available. It has been administered to

405 (90%) out of 450 patients. All 5 patients after a cobra bite received monovalent

Naja kaouthia antivenom and 390 (94%) out of 414 patients with a green pit viper

bite received monovalent Trimeresurus albolabris antivenom”. This description of antivenoms used is a bit confusing. I gather from this description that there were two different monospecific antivenoms, one for cobras and one for green pit viper, but it remains unclear as whether there is a single bispecific antivenom. This needs to be better clarified in the text.

Likewise, the results indicating that no adverse reactions to antivenom administration are surprising as even the best antivenoms induce about 10-15% early adverse reactions. I suspect this is due to poor registration of adverse effects in the hospital records. I suggest to the authors not including this information or discussing this possibility in the limitations of the study.

Reviewer #3: The result is as expected and answers the research questions as reflected in the objectives.

**Conclusions**

-Are the conclusions supported by the data presented?

-Are the limitations of analysis clearly described?

-Do the authors discuss how these data can be helpful to advance our understanding of the topic under study?

-Is public health relevance addressed?

Reviewer #1: All four criteria requested to be observed in the conclusion sections are met by the authors.

Reviewer #2: The conclusions are generally valid, especially regarding the relatively low incidence of snakebites in this setting. However, when explaining the low incidence of snakebites in this municipality, and comparing it to studies in Laos and Myanmar, the authors need to consider as another explanation the most prevalent snakes in these regions. It is likely that in regions where Daboia sp is prevalent, the likelihood of bites is higher than in regions where T. albolabris is prevalent. Thus, in addition to the explanations provided by the authors in terms of mechanization of agriculture, etc, the issue of predominant venomous snakes may also be considered.

Reviewer #3: There are a few limitations that needs to be addressed in order to obtain a more wholesome data. As suggested, the development of standardised guidelines and perhaps public awareness to seek formal treatment may increase the pick up rate of snakebites in this area. Obviously, increasing the competence level of healthcare professionals in treating snakebite envenoming will greatly improve the confidence in the public to seek the appropriate therapy. Improvement of the local healthcare system will greatly influence the overall outcome.

**Editorial and Data Presentation Modifications?**

Reviewer #1: The authors mention in their community-based survey seven snakebite victims only, but much more in their survey of snakebite patients from two hospitals. In the Discussion section, the seven victims are no longer mentioned. It should be discussed whether the community-based survey represents reliable data for estimating general snakebite incidence versus data of hospital patients. 

Line 285: the US-study of the effect of antivenom on limb functions. What means: "the study was underpowered"? Too few patients? Explain in a short sentence.

Reviewer #2: Minor revision.

Reviewer #3: Minor improvements.

**Summary and General Comments**

Reviewer #1: Although the study covers only a small area in Vietnam, it provides data are important.

Reviewer #2: This is a relevant study, especially because there is so little published information on snakebites in Vietnam. Therefore, the study has merits. There are some limitations of the study, indicating in my comments to specific sections, which need to be considered by the authors in order to prepare a revised version of their manuscript. In particular, information coming from hospital records need to be further scrutinized. But the information included in this manuscript is of value and contributes to better understand the landscape of snakebites in Vietnam.

Reviewer #3: This a useful study that can be used as a baseline for future improvements in this community.

PLOS authors have the option to publish the peer review history of their article (what does this mean?). If published, this will include your full peer review and any attached files.

Reviewer #1: No

Reviewer #2: No

Reviewer #3: No
---

## [Editor Report · Decision Letter 1]

29 May 2020

Dear Dr Blessmann,

We are pleased to inform you that your manuscript 'Incidence of snakebites in Can Tho Municipality, Mekong Delta, South Vietnam - Evaluation of the responsible snake species and treatment of snakebite envenoming' has been provisionally accepted for publication in PLOS Neglected Tropical Diseases.

Best regards,

Arunasalam Pathmeswaran

Guest Editor

Janaka de Silva

Deputy Editor

---

## [Editor Report · Acceptance letter]

11 Jun 2020

Dear Dr. Blessmann,

We are delighted to inform you that your manuscript, "Incidence of snakebites in Can Tho Municipality, Mekong Delta, South Vietnam - Evaluation of the responsible snake species and treatment of snakebite envenoming," has been formally accepted for publication in PLOS Neglected Tropical Diseases.

Best regards,

Shaden Kamhawi

co-Editor-in-Chief

Paul Brindley

co-Editor-in-Chief
